# Angular-Accelerometer-Based Flexible-State Estimation and Tracking Controller Design for Hypersonic Flight Vehicle

**Daqiao Zhang [1], Xiaolong Zheng [1], Yangguang Xie [2] and Xiaoxiang Hu [3,*]**

[1] 303 Unit, Xi'an Research Institute of High-Tech, Xi'an 710025, China; jionasaad@gmail.com (D.Z.); nataliiyachica@gmail.com (X.Z.)

[2] Flight Automatic Control Research Institute, Aviation Industry Corporation of China, Xi'an 710065, China; celeenaady@gmail.com

[3] School of Automation, Northwestern Polytechnical University, Xi'an 710072, China

[*] Correspondence: huxiaoxiang2008@gmail.com; Tel.: +86-13669261728

**Abstract:** The controller design of hypersonic flight vehicles is a challenging task, especially when its flexible states are immeasurable. Unfortunately, the flexible states are difficult to measure directly. In this paper, an angular-accelerometer-based method for the estimation of flexible states is proposed. By adding a pitch angel angular accelerometer and designing an Extended Kalman Filter-based online estimation method, the flexible states could be obtained in real time. Then, based on the estimated flexible states, a stable inversion-based controller-design method was utilized, and a robust tracking controller was designed for hypersonic flight vehicles. The proposed method provides an effective means of estimating flexible states and conducting the observer-based controller design of hypersonic flight vehicles. Finally, a numeral simulation is given to show the effectiveness of the proposed control method.

**Keywords:** hypersonic flight vehicle (HFV); immeasurable flexible states; Extended Kalman Filter



## 1. Introduction

The hypersonic flight vehicle (HFV) is a type of vehicle that has the ability of horizontal launching and multiple uses, and thus, it is widely studied nowadays. For severe maneuver demands, the air-breathing scramjet engines of HFVs are integrated with the airframe systems [1]. This makes the couplings and flight dynamics of HFVs very complex. The controller design of HFVs is a challenging task and has been a hot topic in recent years. The complex nonlinear model proposed in [2] has been adopted by researchers. The classical linear model-based controller-designed method has been applied in HFVs [3]. However, the changing speeds of their flight dynamics are very high, so linear models cannot exactly control the complex dynamics of HFVs. In such cases, a nonlinear controller is needed for HFVs. Existing nonlinear control methods have also been considered, such as the T-S fuzzy controller [4], the neural network-based controller [5,6], the back-stepping controller [7,8], the nonlinear adaptive controller [9,10], the adaptive dynamic programming controller [11]. However, in the application of the above nonlinear controllers, the states of HFV are all assumed to be measurable, while in most situations, some states of a real system are immeasurable, especially for flight vehicles.

As a simple and common solution, the acceleration of a flight vehicle can be easily measured by an accelerometer, and the angular flight velocity of a flight vehicle can also be tested by a gyroscope. In this case, the flight velocity and attitude can be computed by means of the integration of the acceleration, and the flight angle of the flight vehicle can be computed by the measurable angular velocity. However, the flexible dynamics of a flight vehicle are difficult to measure, since the vibration is not at a fixed point. In these cases, "all states are measurable" is not an appropriate assumption, and the observer design problem of HFVs must be considered.

Observer-based controller-design methods for HFVs have also been studied; in [12], the angle of attack and the flight path angle are assumed to be immeasurable and a high-order sliding mode observer is designed, and in [13], the coupling between rigid and flexible dynamics is assumed to be unknown, and the flexible dynamics of the HFV is viewed as disturbance, then an observer is constructed to estimate it. Most of the observer-based controller-design methods for HFVs assume the nonlinear disturbance is unknown and then obtain the disturbance by designing an observer. Just as we have illustrated, the angle of attack and the flight path angle can be measurable using a gyroscope or can be computed, so this assumption is not suitable for a real flight vehicle. The couplings between rigid and flexible dynamics are objective and can be modeled by flight tests. The observer disturbance is really needed for HFVs since their flight environments are really complex. However, the flexible dynamics of HFV are hard to measure in practice, while few results can be found about this topic.

Considering the flexible dynamics, the model of HFVs is a nonlinear non-minimum-phase model, and the flexible dynamics of HFVs are all internal dynamics [14]. Observer design for internal dynamics is difficult for non-minimum-phase systems since they do not appear directly in the output of the plant. In this case, an "intermediate variable", which includes the information of the internal dynamics and can easily be measured, is needed for the controller design. For a flight vehicle, the choice of the "intermediate variable" is very difficult since the measuring device of this "intermediate variable" should also be considered.

On the basis of the above discussions, we propose an angular-accelerometer-based flexible-state estimation method and a consequent nonlinear-tracking-controller-design method for HFVs in this paper. The nonlinear dynamics of HIV are reviewed, together with the internal dynamics. Then, an analysis of the flexible dynamics of HFV is presented. Based on the coupling of angular velocity and flexible dynamics, an angular-accelerometer-based flexible-state estimation method is proposed. Since the Extended Kalman Filter (EKF) is a well-known method for state estimation [15,16], it is utilized here for the estimation of flexibility. After obtaining the estimation of flexible states, an ideal internal dynamics (IID) based nonlinear stable inversion controller was utilized for the controller design of HFVs [17]. After obtaining the IID, the output tracking control objective could then be transformed into state tracking, then an inversion-based controller was designed more easily. Based on the constructed state-tracking mode, the computed IID and the estimation results of flexible states, an observer-based nonlinear controller was designed. Simulations of the nonlinear model of HFVs are given to check the correctness of the proposed method.

The main contributions of the paper are summarized as follows:

(1) An angular-accelerometer-based estimation method is proposed for the estimation of immeasurable flexible states of HFVs. Using an angular accelerometer, the flexible states of HFVs could be estimated in real time;

(2) An observer-based nonlinear controller was constructed for the output tracking control of non-minimum-phase systems. The proposed method can be utilized not only for HFVs, but also other non-minimum-phase flight vehicles.

## 2. Problem Formulation

### 2.1. Nonlinear Dynamics of HFVs

The nonlinear dynamics of HFVs is a complete model that was built in [1]; a longitudinal sketch of HFVs is given in Figure 1 and the details are listed as follows:

$$
\begin{aligned}
\dot{h} &= V\sin(\theta - \alpha) \\
\dot{V} &= \frac{1}{m}(T\cos\alpha - D) - g\sin(\theta - \alpha) \\
\dot{\alpha} &= \frac{1}{mV}(-T\sin\alpha - L) + Q + \frac{g}{V}\cos(\theta - \alpha) \\
\dot{\theta} &= Q \\
\dot{Q} &= \frac{M}{I_{yy}} + \frac{\psi_1\ddot{\eta}_1}{I_{yy}} + \frac{\psi_2\ddot{\eta}_2}{I_{yy}} \\
\ddot{\eta}_1 &= -2\varsigma_1\omega_1\dot{\eta}_1 - \omega_1^2\eta_1 + N_1 - \frac{\psi_1 M}{I_{yy}} - \frac{\psi_1\psi_2\ddot{\eta}_2}{I_{yy}} \\
\ddot{\eta}_2 &= -2\varsigma_2\omega_2\dot{\eta}_2 - \omega_2^2\eta_2 + N_2 - \frac{\psi_2 M}{I_{yy}} - \frac{\psi_1\psi_2\ddot{\eta}_1}{I_{yy}}
\end{aligned}
\tag{1}
$$

where $h$ is the flight altitude of HFVs, $V$ is the flight velocity, $\alpha$ is angle of attack and $\theta$ is flight pitch angle. $Q$ is the pitch rate, $M$ is the pitching moment, $\eta_i(i=1,2)$ denotes the generalized elastic coordinate. $\psi_1$ and $\psi_2$ are constrained beam-coupling constants for $\eta_1$ and $\eta_2$. Since $\eta_i(i=1,2)$ is the flexible dynamics of the HFV, (1) has five rigid-body state variables and four flexible ones.

$$
\begin{aligned}
L &\approx \tfrac{1}{2}\rho V^2 S C_L(\alpha, \delta_e) \\
D &\approx \tfrac{1}{2}\rho V^2 S C_D(\alpha, \delta_e) \\
M &\approx z_T T + \tfrac{1}{2}\rho V^2 S\bar{c}(C_{M,\alpha}(\alpha) + C_{M,\delta_e}(\delta_e)) \\
T &\approx C_T^{\alpha^3}\alpha^3 + C_T^{\alpha^2}\alpha^2 + C_T^{\alpha}\alpha + C_T^0 \\
N_1 &\approx N_1^{\alpha^2}\alpha^2 + N_1^{\alpha}\alpha + N_1^0 \\
N_2 &\approx N_2^{\alpha^2}\alpha^2 + N_2^{\alpha}\alpha + N_2^{\delta_e}\delta_e + N_2^0
\end{aligned}
\tag{2}
$$

and

$$
\begin{aligned}
\rho &= \rho_0 \exp(\tfrac{-(h-h_0)}{h_s}) \\
C_L &= C_L^{\alpha}\alpha + C_L^{\delta_e}\delta_e + C_L^0 \\
C_D &= C_D^{\alpha^2}\alpha^2 + C_D^{\alpha}\alpha + C_D^{\delta_e^2}\delta_e^2 + C_D^{\delta_e}\delta_e + C_D^0 \\
C_{M,\alpha} &= C_{M,\alpha}^{\alpha^2}\alpha^2 + C_{M,\alpha}^{\alpha}\alpha + C_{M,\alpha}^0 \\
C_{M,\delta_e} &= c_e\delta_e, \\
\bar{q} &= \tfrac{1}{2}\rho V^2 \\
C_T^{\alpha^3} &= \beta_1\Phi + \beta_2 \\
C_T^{\alpha^2} &= \beta_3\Phi + \beta_4 \\
C_T^{\alpha} &= \beta_5\Phi + \beta_6 \\
C_T^0 &= \beta_7\Phi + \beta_8.
\end{aligned}
\tag{3}
$$

where $C_D$ is the drag coefficient; $C_L$ is the lift coefficient; $C_{M,\alpha}$ is the drag coefficient; $C_{M,i}$ is the contribution to the moment, $C_T^{\alpha_i}$ is the $i$th order coefficient of $\alpha$ in $T$, $\beta_i$ is the $i$th thrust fit parameter. $u = [\Phi, \delta_e]^T$ is the input of the plant.

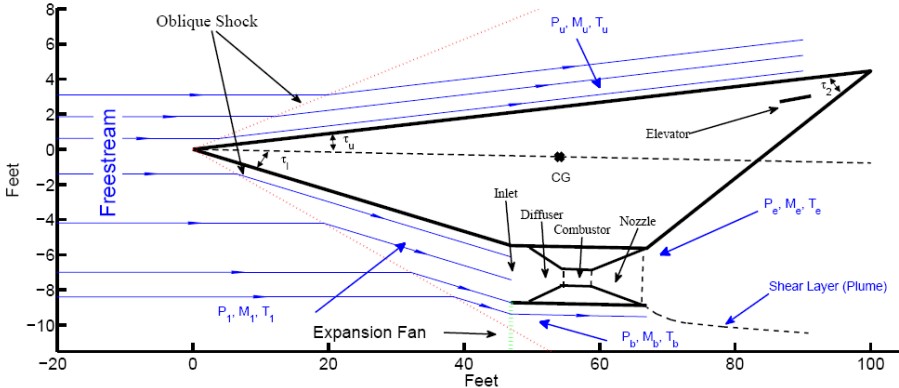

**Figure 1.** Geometry of HFV model.

**Remark 1.** *The nonlinear model considered in this paper is far more complex than the one adopted in most studies in the literature, since the coupling between the angular acceleration $\dot{Q}$ and the flexible states $[\eta_1, \dot{\eta}_1, \eta_2, \dot{\eta}_2]^T$ is often neglected, which means that, in most studies, $\dot{Q} = \frac{M}{I_{yy}}$ is adopted for the system analysis and control design instead of $\dot{Q} = \frac{M}{I_{yy}} + \frac{\psi_1\ddot{\eta}_1}{I_{yy}} + \frac{\psi_2\ddot{\eta}_2}{I_{yy}}$.*

### 2.2. Input/Output Linearization

By combining (2) and (3) into (1), the nonlinear model of HFVs is transformed into an affine form, and is rewritten as:

$$
\begin{aligned}
\dot{h} &= V\sin(\theta - \alpha) \\
\dot{V} &= \frac{1}{m}(T\cos\alpha - D) - g\sin(\theta - \alpha) \\
&= \frac{1}{m}\left[\beta_2\alpha^3 + \beta_4\alpha^2 + \beta_6\alpha + \beta_8\right]\cos\alpha \\
&\quad - \frac{1}{m}\left[\frac{1}{2}\rho V^2 S\left(C_D^{\alpha^2}\alpha^2 + C_D^{\alpha}\alpha + C_D^0\right)\right] - g\sin(\theta - \alpha) \\
&\quad + \frac{1}{m}\left\{\left[\beta_1\alpha^3 + \beta_2\alpha^2 + \beta_5\alpha + \beta_7\right]\cos\alpha\right\}\Phi \\
&\quad - \frac{1}{m}\left[\frac{1}{2}\rho V^2 S\left(C_D^{\delta_e^2}\delta_e + C_D^{\delta_e}\right)\right]\delta_e \\[4pt]
\dot{\alpha} &= -\frac{1}{mV}\left[\beta_2\alpha^3 + \beta_4\alpha^2 + \beta_6\alpha + \beta_8\right]\sin\alpha \\
&\quad - \frac{1}{mV}\left[\frac{1}{2}\rho V^2 S\left(C_L^{\alpha}\alpha + C_L^0\right)\right] + Q + \frac{g}{V}\cos(\theta - \alpha) \\
&\quad - \frac{1}{mV}\left\{\left[\beta_1\alpha^3 + \beta_3\alpha^2 + \beta_5\alpha + \beta_7\right]\sin\alpha\right\}\Phi \\
&\quad - \frac{1}{mV}\left[\frac{1}{2}\rho V^2 S C_L^{\delta_e}\right]\delta_e \\[4pt]
\dot{\theta} &= Q \\
\dot{Q} &= \frac{1}{I_{yy}}\left\{z_T\left[\beta_2\alpha^3 + \beta_4\alpha^2 + \beta_6\alpha + \beta_8\right]\right. \\
&\quad \left. + \frac{1}{2}\rho V^2 S\bar{c}\left(C_{M,\alpha}^{\alpha^2}\alpha^2 + C_{M,\alpha}^{\alpha}\alpha + C_{M,\alpha}^0\right)\right\} \\
&\quad + \frac{1}{I_{yy}}\left\{z_T\left[\beta_1\alpha^3 + \beta_3\alpha^2 + \beta_5\alpha + \beta_7\right]\right\}\Phi \\
&\quad + \frac{1}{I_{yy}}\left\{\frac{1}{2}\rho V^2 S\bar{c}c_e\right\}\delta_e + \frac{\psi_1\ddot{\eta}_1}{I_{yy}} + \frac{\psi_2\ddot{\eta}_2}{\%I_{yy}}\% \\[4pt]
\ddot{\eta}_1 &= -2\varsigma_1\omega_1\dot{\eta}_1 - \omega_1^2\eta_1 + N_1^{\alpha^2}\alpha^2 + N_1^{\alpha}\alpha + N_1^0 - \frac{\psi_1 M}{I_{yy}} - \frac{\psi_1\psi_2\ddot{\eta}_2}{I_{yy}} \\
\ddot{\eta}_2 &= -2\varsigma_2\omega_2\dot{\eta}_2 - \omega_2^2\eta_2 + N_2^{\alpha^2}\alpha^2 + N_2^{\alpha}\alpha + N_2^0 + N_2^{\delta_e}\delta_e - \frac{\psi_2 M}{I_{yy}} - \frac{\psi_1\psi_2\ddot{\eta}_1}{I_{yy}}.
\end{aligned}
$$

Considering the form of the affine nonlinear system, the above equation can be rewritten as:

$$\dot{x} = f(x) + g(x)u \tag{4}$$

where $x = [h, V, \alpha, \theta, Q, \eta_1, \dot{\eta}_1, \eta_2, \dot{\eta}\%_2]^T$, $u = [\Phi, \delta_e]^T$, and

$$
f(x) = \begin{bmatrix} f_1(x) & f_2(x) & f_3(x) & f_4(x) & f_5(x) & f_6(x) & f_7(x) & f_8(x) & f_9(x) \end{bmatrix}^T
$$

$$
g(x(t)) = \begin{bmatrix} 0 & b_{21} & b_{31} & 0 & b_{51} & 0 & b_{71} & 0 & b_{91} \\ 0 & b_{22} & b_{32} & 0 & b_{52} & 0 & b_{72} & 0 & b_{92} \end{bmatrix}^T,
$$

$$
\begin{aligned}
f_1(x) &= V\sin(\theta - \alpha) \\
f_2(x) &= \frac{1}{m}\left[\beta_2\alpha^3 + \beta_4\alpha^2 + \beta_6\alpha + \beta_8\right]\cos\alpha - \frac{1}{m}\left[\frac{1}{2}\rho V^2 S\left(C_D^{\alpha^2}\alpha^2 + C_D^{\alpha}\alpha + C_D^0\right)\right] - g\sin(\theta - \alpha) \\
f_3(x) &= -\frac{1}{mV}\left[\beta_2\alpha^3 + \beta_4\alpha^2 + \beta_6\alpha + \beta_8\right]\sin\alpha - \frac{1}{mV}\left[\frac{1}{2}\rho V^2 S\left(C_L^{\alpha}\alpha + C_L^0\right)\right] + Q + \frac{g}{V}\cos(\theta - \alpha) \\
f_4(x) &= Q \\
f_5(x) &= \frac{1}{I_{yy}}\left\{z_T\left[\beta_2\alpha^3 + \beta_4\alpha^2 + \beta_6\alpha + \beta_8\right] + \frac{1}{2}\rho V^2 S\bar{c}\left(C_{M,\alpha}^{\alpha^2}\alpha^2 + C_{M,\alpha}^{\alpha}\alpha + C_{M,\alpha}^0\right)\right\} + \frac{\psi_1 f_7(x)}{I_{yy}} + \frac{\psi_2 f_9(x)}{I_{yy}} \\[4pt]
f_6(x) &= \dot{\eta}_1 \\
f_7(x) &= -2\varsigma_1\omega_1\dot{\eta}_1 - \omega_1^2\eta_1 + \left(N_1^{\alpha^2} - \frac{\psi_1\psi_2}{I_{yy}k_2}N_2^{\alpha^2}\right)\alpha^2 + \left(N_1^{\alpha} - \frac{\psi_1\psi_2}{I_{yy}k_2}N_2^{\alpha}\right)\alpha \\
&\quad + N_1^0 - \frac{\psi_1\psi_2}{I_{yy}k_2}N_2^0 + \frac{\psi_1\psi_2}{I_{yy}k_2}2\varsigma_2\omega_2\dot{\eta}_2 + \frac{\psi_1\psi_2}{I_{yy}k_2}\omega_2^2\eta_2 \\
&\quad - \frac{\psi_1 I_{yy}k_2 - \psi_1\psi_2^2}{I_{yy}^2 k_2}\left(z_T\left[\beta_2\alpha^3 + \beta_4\alpha^2 + \beta_6\alpha + \beta_8\right] + \frac{1}{2}\rho V^2 S\bar{c}\left(C_{M,\alpha}^{\alpha^2}\alpha^2 + C_{M,\alpha}^{\alpha}\alpha + C_{M,\alpha}^0\right)\right)
\end{aligned}
$$

$$f_8(x) = \dot{\eta}_2$$
$$f_9(x) = -2\varsigma_2\omega_2\dot{\eta}_2 - \omega_2^2\eta_2 + \left(N_2^{\alpha^2} - \frac{\psi_1\psi_2}{I_{yy}k_1}N_1^{\alpha^2}\right)\alpha^2 + \left(N_2^{\alpha} - \%\frac{\psi_1\psi_2}{I_{yy}k_1}N_1^{\alpha}\right)\alpha$$
$$+N_2^0 - \frac{\psi_1\psi_2}{I_{yy}k_1}N_2^0 + \frac{\psi_1\psi_2}{I_{yy}k_1}2\varsigma_1\omega_1\dot{\eta}_1 + \frac{\psi_1\psi_2}{I_{yy}k_1}\omega_1^2\eta_1$$
$$-\frac{\psi_2 I_{yy}k_1 - \psi_2\psi_1^2}{I_{yy}^2k_1}\left(z_T\left[\beta_2\alpha^3 + \beta_4\alpha^2 + \beta_6\alpha + \beta_8\right] + \frac{1}{2}\rho V^2 S\bar{c}\left(C_{M,\alpha}^{\alpha^2}\alpha^2 + C_{M,\alpha}^{\alpha}\alpha + C_{M,\alpha}^0\right)\right)$$

$$b_{21} = \frac{1}{m}\left\{\left[\beta_1\alpha^3 + \beta_3\alpha^2 + \beta_5\alpha + \beta_7\right]\cos\alpha\right\}$$
$$b_{22} = -\frac{1}{m}\left[\frac{1}{2}\rho V^2 S\left(C_D^{\delta_e^2}\delta_e + C_D^{\delta_e}\right)\right]$$
$$b_{31} = -\frac{1}{mV}\left\{\left[\beta_1\alpha^3 + \beta_3\alpha^2 + \beta_5\alpha + \beta_7\right]\sin\alpha\right\}$$
$$b_{32} = -\frac{1}{mV}\left[\frac{1}{2}\rho V^2 S C_L^{\delta_e}\%\right]$$
$$b_{51} = \frac{1}{I_{yy}}\left\{z_T\left[\beta_1\alpha^3 + \beta_3\alpha^2 + \beta_5\alpha + \beta_7\right]\right\}$$
$$b_{52} = \frac{1}{I_{yy}}\left\{\frac{1}{2}\rho V^2 S\bar{c}c_e\right\}$$
$$b_{71} = -\frac{\psi_1 I_{yy}k_2 - \psi_1\psi_2^2}{I_{yy}^2k_2\%}z_T\left[\beta_1\alpha^3 + \beta_3\alpha^2 + \beta_5\alpha + \beta_7\right]$$
$$b_{72} = -\frac{\psi_1 I_{yy}k_2 - \psi_1\psi_2^2}{I_{yy}^2k_2\%}\frac{1}{2}\rho V^2 S\bar{c}c_e - \frac{\psi_1\psi_2}{I_{yy}k_2}N_2^{\delta_e}$$
$$b_{91} = -\frac{\psi_2 I_{yy}k_1 - \psi_2\psi_1^2}{I_{yy}^2k_1\%}z_T\left[\beta_1\alpha^3 + \beta_3\alpha^2 + \beta_5\alpha + \beta_7\right]$$
$$b_{92} = -\frac{\psi_2 I_{yy}k_1 - \psi_2\psi_1^2}{I_{yy}^2k_1\%}\frac{1}{2}\rho V^2 S\bar{c}c_e + N_2^{\delta_e}.$$

Choosing the output of HFVs as $y(t) = [h, V]^T$, and following the input–output linearization method, the relative degree of (4) can be computed and $r = [2, 1]$, so the nonlinear model (4) can only be partially linearized, and the partially linearized model is:

$$\dot{h} = V\sin(\theta - \alpha)$$
$$\ddot{h} = L_f^2 h + \begin{bmatrix} L_{g1}L_f h & L_{g2}L_f h \end{bmatrix}u \qquad (5)$$
$$\dot{V} = f_2(x) + \begin{bmatrix} b_{21}(x) & b_{22}(x) \end{bmatrix}u,$$

where

$$L_f^2 h = \sin(\theta - \alpha)\left\{\frac{1}{m}\left[\beta_2\alpha^3 + \beta_4\alpha^2 + \beta_6\alpha + \beta_8\right]\cos\alpha - \frac{1}{m}\left[\frac{1}{2}\rho V^2 S\left(C_D^{\alpha^2}\alpha^2 + C_D^\alpha\alpha + C_D^0\right)\right] - g\sin(\theta - \alpha)\right\}$$
$$-V\cos(\theta - \alpha)\left\{-\frac{1}{mV}\left[\beta_2\alpha^3 + \beta_4\alpha^2 + \beta_6\alpha + \beta_8\right]\sin\alpha + \frac{1}{mV}\left[\frac{1}{2}\rho V^2 S\left(C_L^\alpha\alpha + C_L^0\right)\right] + \frac{g}{V}\cos(\theta - \alpha)\right\}$$
$$L_{g1}L_f h = \sin(\theta - \alpha)\frac{1}{m}\left\{\left[\beta_1\alpha^3 + \beta_3\alpha^2 + \beta_5\alpha + \beta_7\right]\cos\alpha\right\} + V\cos(\theta - \alpha)\frac{1}{mV}\left\{\left[\beta_1\alpha^3 + \beta_3\alpha^2 + \beta_5\alpha + \beta_7\right]\sin\alpha\right\}$$
$$L_{g2}L_f h = -\sin(\theta - \alpha)\frac{1}{m}\left[\frac{1}{2}\rho V^2 S\left(C_D^{\delta_e^2}\delta_e + C_D^{\delta_e}\right)\right] + V\cos(\theta - \alpha)\frac{1}{mV}\left[\frac{1}{2}\rho V^2 S C_L^{\delta_e}\right].$$

Defining $\zeta = [h, \dot{h}, V]^T$ as the external state, then (5) is

$$\dot{\zeta} = A_\zeta\zeta + B_\zeta\left[F_\zeta(x) + G_\zeta(x)u\right] \qquad (6)$$

where

$$A_\zeta = \begin{bmatrix} 0 & 1 & 0 \\ 0 & 0 & 0 \\ 0 & 0 & 0 \end{bmatrix}, B_\zeta = \begin{bmatrix} 0 & 0 \\ 1 & 0 \\ 0 & 1 \end{bmatrix}, F_\zeta(x) = \begin{bmatrix} L_f^2 h \\ f_2(x) \end{bmatrix},$$
$$G_\zeta(x) = \begin{bmatrix} L_{g1}L_f h & L_{g2}L_f h \\ g_{21}(x) & g_{22}(x) \end{bmatrix}.$$

For the analysis of the internal dynamics, a virtual controller is introduced

$$v = F_\zeta(x) + G_\zeta(x)u$$

so

$$u = G_\zeta^{-1}(x)\left[v - F_\zeta(x)\right]$$

Choosing the internal states as $\varsigma = [\alpha, Q, \eta_1, \dot{\eta}_1, \eta_2, \dot{\eta}_2]^T$, the internal dynamics of (1) can be constructed, and

$$
\begin{aligned}
\dot{\varsigma} &=
\begin{bmatrix}
f_3(x) \\
f_5(x) \\
f_6(x) \\
f_7(x) \\
f_8(x) \\
f_9(x)
\end{bmatrix}
+
\begin{bmatrix}
b_{31} & b_{32} \\
b_{51} & b_{52} \\
0 & 0 \\
b_{71} & b_{72} \\
0 & 0 \\
b_{91} & b_{92}
\end{bmatrix}
G_{\xi}^{-1}(x)\big[v - F_{\xi}(x)\big] \\
&= F_{\varsigma}(x) + G_{\varsigma}(x)\big[F_{\xi}(x) + G_{\xi}(x)u\big],
\end{aligned}
\tag{7}
$$

where

$$
F_{\varsigma}(x) =
\begin{bmatrix}
f_3(x) \\
f_5(x) \\
f_6(x) \\
f_7(x) \\
f_8(x) \\
f_9(x)
\end{bmatrix}
-
\begin{bmatrix}
b_{31} & b_{32} \\
b_{51} & b_{52} \\
0 & 0 \\
b_{71} & b_{72} \\
0 & 0 \\
b_{91} & b_{92}
\end{bmatrix}
G_{\xi}^{-1}(x) F_{\xi}(x)
$$

$$
G_{\varsigma}(x) =
\begin{bmatrix}
b_{31} & b_{32} \\
b_{51} & b_{52} \\
0 & 0 \\
b_{71} & b_{72} \\
0 & 0 \\
b_{91} & b_{92}
\end{bmatrix}
G_{\xi}^{-1}(x).
$$

Then, there exists a diffeomorphism coordinate transformation, $x \to (\ \xi^T, \ \varsigma^T\ )^T$, and the nonlinear model of HFV (1) can be transformed into (6) and (7). Utilizing the method of analysis proposed in [10], the internal dynamics (7) is of the non-minimum-phase type.

**Remark 2.** *In the nonlinear expression of (7), x is utilized to express the nonlinear dynamics rather than* $(\xi^T, \ \varsigma^T\ )^T$*. Because the diffeomorphism coordinate transformation adopted here is simple, for the convenience of reading and understanding, the symbol x is still utilized to express it.*

Though expressions (6) and (7) are complex, the nonlinear model is greatly simplified. After the transformation, also considering the fact that the flexible states $[\eta_1, \dot{\eta}_1, \eta_2, \dot{\eta}_2]$ are immeasurable, the control objective is converted into: finding an observer for the flexible state $[\eta_1, \dot{\eta}_1, \eta_2, \dot{\eta}_2]$, then designing an observer-based controller $u$ for system (6) and (7). Since the nonlinearity of the original system (1) is greatly reduced, the design of the controller for (6) and (7) is far easier than the one for (1).

## 3. Estimation of Flexible Dynamics

For the state-feedback controller, all states should be measurable. Unfortunately, in most cases, some states of real systems are immeasurable. For HFV, the acceleration can be measured by an accelerometer, and the pitch rate $Q$ can be tested by a gyroscope; then, $V$ and $h$ can be computed via the integration of the acceleration, and $\alpha$ can be computed via the integration of $Q$, so $[h, \dot{h}, V, \alpha, Q]^T$ can be viewed as a measurable state. However, the flexible dynamics $\eta = [\eta_1, \dot{\eta}_1, \eta_2, \dot{\eta}_2]^T$ are difficult measure, since the vibration is not at a fixed point, it is a vibration of the whole airframe. Furthermore, the flexible dynamics contain the first-order vibration, $\dot{\eta}_1, \dot{\eta}_2$, and they are very difficult to be exactly measured. In this case, an alternative scheme should be considered.

Observing the nonlinear model of HFVs, there is coupling between the angular acceleration $\dot{Q}$ and flexible states $\eta$. The measurement of an angular acceleration is a commonly used approach and has been widely used in the guidance and navigation of flight vehicles, and it can be easily accomplished using an angular accelerometer. Furthermore, an angular

accelerometer is easy to fix, and its measured value is trustworthy. If the flexible states can be estimated by the coupling between angular acceleration and flexible states, the controller design can be easily carried out for HFVs. Thus, in this study, an angular-accelerometer-based flexible dynamics estimations strategy was utilized and a corresponding estimation method was built.

### 3.1. Extend Kalman Filter

From (1), angular acceleration $\dot{Q} = \frac{M}{I_{yy}} + \frac{\psi_1 \ddot{\eta}_1}{I_{yy}} + \frac{\psi_2 \ddot{\eta}_2}{I_{yy}}$, where $I_{yy}$ is already known and $M = z_T T + \frac{1}{2}\rho V^2 S(C_{M,\alpha}(\alpha) + C_{M,\delta_e}(\delta_e))$ can beastly computed by the measurable states, if angular acceleration $\dot{Q}$ can be measured in real time, we can deem that, $\frac{\psi_1 \ddot{\eta}_1}{I_{yy}} + \frac{\psi_2 \ddot{\eta}_2}{I_{yy}}$ can be measured in real time. Defining a new variable $\eta_y$, and $\eta_y = \frac{\psi_1}{I_{yy}}\ddot{\eta}_1 + \frac{\psi_2}{I_{yy}}\ddot{\eta}_2 = \dot{Q} - \frac{M}{I_{yy}}$, since $\eta = [\eta_1, \dot{\eta}_1, \eta_2, \dot{\eta}_2]^T$ is unmeasurable, a estimation method is needed here to estimate $\eta$ based on the measurable value of $\eta_y$.

Because of the vibration and wind disturbance of HFV, when utilizing an angular accelerometer, the measured angular acceleration $\dot{Q}$ will suffer from stochastic disturbances, so $\eta_y$ can be rewrote as

$$\eta_y = \frac{\psi_1}{I_{yy}}\ddot{\eta}_1 + \frac{\psi_2}{I_{yy}}\ddot{\eta}_2 + \vartheta(t) = \Gamma(x) + \vartheta(t) \tag{8}$$

where $\Gamma(x) = \frac{\psi_1}{I_{yy}}\ddot{\eta}_1 + \frac{\psi_2}{I_{yy}}\ddot{\eta}_2$, $\vartheta(t)$ is the measurement noise and is assumed to be a zero-mean Gaussian white process. Similarly, the flexible dynamics of HFVs are also subjected to stochastic disturbances, and then

$$\dot{\eta} = \begin{bmatrix} f_6(x) \\ f_7(x) \\ f_8(x) \\ f_9(x) \end{bmatrix} + \begin{bmatrix} 0 & 0 \\ b_{71} & b_{72} \\ 0 & 0 \\ b_{91} & b_{92} \end{bmatrix} u + \varpi(t) = F_\eta(x) + G_\eta(x)u + \varpi(t)$$

where $\varpi(t)$ represents the stochastic disturbances. Then, an estimation method is needed for the immeasurable flexible state $\eta$ on the basis of the measurable value of $\eta_y$.

The nonlinear dynamics of flexible dynamics are really complex, and furthermore, they suffer from stochastic disturbances. In this case, a simple linear observer was not suitable here. EKF can be utilized for nonlinear systems with parameter variations, measurement noise and system uncertainties, so EKF is an efficient means of optimally estimating nonlinear processes and has been widely used. In this case, the EKF method was utilized for the flexible-dynamics estimations of HFVs.

According to the EKF approach, the observer for flexible states $\eta$ is constructed as

$$\begin{aligned} \dot{\hat{\eta}} &= A_\eta(t)\hat{\eta} + B_\eta(t)u + L_\eta(t)(\eta_y - C_\eta(t)\hat{\eta}) + \varpi(t) \\ \eta_y &= C_\eta(t)\hat{\eta} + \vartheta(t) \end{aligned} \tag{9}$$

where $\hat{\eta}$ is the observer state, and $L_\eta(t)$ is the observer gain, which will be designed later;

$$A_\eta(t) = \frac{\partial F_\eta(x)}{\partial \eta}, B_\eta(t) = \frac{\partial G_\eta(x)}{\partial \eta}, C_\eta(t) = \frac{\partial \Gamma(x)}{\partial \eta}$$

then the estimation error system can be easily constructed:

$$\dot{e}_{\hat{\eta}} = A_\eta(t)e_{\hat{\eta}} - L_\eta(t)C_\eta(t)e \tag{10}$$

where $e_{\hat{\eta}} = \eta - \hat{\eta}$ is the estimation error.

Assuming that $(A_\eta(t), C_\eta(t))$ of (9) is all observable, the EKF gain can be computed as

$$L_\eta(t) = P_\eta(t)C_\eta(t)R_\eta^{-1}(t)$$

where $P_\eta(t)$ is a symmetric and positive definite value, and is the solution of the following Riccati equation

$$\dot{P}_\eta(t) = A_\eta(t)P_\eta(t) + P_\eta(t)A_\eta^T(t) + Q_\eta(t) - P_\eta(t)C_\eta^T(t)R^{-1}(t)C_\eta(t)R_\eta^{-1}(t)P_\eta(t)$$
$$P_\eta(t) = P_\eta(0) > 0$$

where $Q_\eta(t)$ and $R_\eta(t)$ are positive definite matrices.

Since a numerical solution can be easily obtained for the above algebraic Riccati equation, based on the calculated observer gain $L_\eta(t)$, the convergence of the flexible states observer error system (10) can be guaranteed, then $\eta$ can be estimated in real time.

### 3.2. $\hat{\eta}$ Based Control Objective

Though the complexity of the nonlinear expression of HFV was greatly reduced by input–output linearization, it was still difficult to conduct controller design directly. For the carrying out of the controller design, nonlinear expression (8) was linearized first according to the trim point of HFVs.

$$\dot{\varsigma} = A_{\varsigma 1}\xi + A_{\varsigma 2}\varsigma + B_\varsigma v$$
$$A_{\varsigma 1} = \frac{\partial F_\varsigma(x)}{\partial \xi}\Big|_{\xi=\xi_0,\varsigma=\varsigma_0,v=0}, \quad A_{\varsigma 2} = \frac{\partial F_\varsigma(x)}{\partial \varsigma}\Big|_{\xi=\xi_0,\varsigma=\varsigma_0,v=0},$$
$$B_\varsigma = \frac{\partial G_\varsigma(x)}{\partial v}\Big|_{\xi=\xi_0,\varsigma=\varsigma_0,v=0},$$

where $\xi = \xi_0$, $\varsigma = \varsigma_0$ is the trim point of HFVs.

Considering the control objective and the non-minimum-phase characteristics of HFVs, the stable inversion method was adopted here. IID is an ideal trajectory of internal dynamics, which is drawn by the given reference trajectory. Combining the flight task of HFV, reference trajectory of HFV is $y^d = [h^d, V^d]^T$, and IID is $\varsigma^d = [\alpha^d, Q^d, \eta_1^d, \dot{\eta}_1^d, \eta_2^d, \dot{\eta}_2^d]^T$. IID can be computed easily [12], and then the original problem can be transformed into a new one:

$$\dot{e} = Ae + B\left[F_\xi(x) + G_\xi(x)u - y^{d(r)}\right] \tag{11}$$

where $e = [e_\xi^T, e_\varsigma^T]^T$, $e_\xi = \xi - \xi^d$, $e_\varsigma = \varsigma - \varsigma^d$, $\xi^d = \begin{bmatrix} h^d, & \dot{h}^d, & V^d \end{bmatrix}^T$ is already given and $y^{d(r)} = \begin{bmatrix} \ddot{h}^d, & \dot{V}^d \end{bmatrix}^T$ represents the derivatives of the given reference trajectory,

$$A = \begin{bmatrix} A_\xi & 0 \\ A_{\varsigma 1} & A_{\varsigma 2} \end{bmatrix}, B = \begin{bmatrix} B_\xi \\ B_\varsigma \end{bmatrix}$$

For the controller design, introducing a new segmentation of the error state, $e = [e_\chi^T, e_\eta^T]^T$, $e_\chi = \chi - \chi^d$, $e_\eta = \eta - \eta^d$, where $\chi = \begin{bmatrix} h, & \dot{h}, & V, & \alpha & Q \end{bmatrix}^T$ represents the measurable states, and $\eta = [\eta_1, \dot{\eta}_1, \eta_2, \dot{\eta}_2]^T$ represents the immeasurable flexible states. Considering the estimated error system (10), the closed loop system under the original control objective is given by

$$\dot{e} = Ae + B\left[F_\xi(x) + G_\xi(x)u - y^{d(r)}\right],$$
$$\dot{e}_{\hat{\eta}} = (A_\eta(t) - L_\eta(t)C_\eta(t))e_{\hat{\eta}} \tag{12}$$

with the diffeomorphism coordinate transformation, $x \rightarrow (\begin{array}{cc} \xi^T, & \varsigma^T \end{array})^T \rightarrow (\begin{array}{cc} \chi^T, & \eta^T \end{array})^T$. Then, the control objective is as follows: for the state-tracking error system together with the

estimation error (12), design an observer-based controller $u = \psi(\chi, \hat{\eta})$, then the closed-loop system (12) is stable.

**Remark 3.** *In (12), $e = [e_\chi^T, e_\eta^T]^T$ is introduced for error state $e$. Since the rigid-body-state error $e_\chi$ is measurable, and the flexible-state error $e_\eta$ is immeasurable, the new segmentation is convenient for our controller design.*

## 4. Controller Design

For the observer error system (10), a Lyapunov function is designed as: $V_{\hat{\eta}} = e_{\hat{\eta}}^T P_\eta^{-1}(t) e_{\hat{\eta}}$, where $P_\eta(t)$ is a symmetric and positive definite matrix. Then,

$$
\begin{aligned}
\dot{V}_{\hat{\eta}} &= \dot{e}_{\hat{\eta}}^T P_\eta^{-1}(t) e_{\hat{\eta}} + e_{\hat{\eta}}^T P_\eta^{-1}(t) \dot{e}_{\hat{\eta}} \\
&= e_{\hat{\eta}}^T \big(A_\eta(t) - L_\eta(t) C_\eta(t)\big)^T P_\eta^{-1}(t) e_{\hat{\eta}} + e_{\hat{\eta}}^T P_\eta^{-1}(t) \big(A_\eta(t) - L_\eta(t) C_\eta(t)\big) e_{\hat{\eta}} \\
&\leq -e_{\hat{\eta}}^T P_\eta^{-1}(t) \Big[ Q_\eta(t) + P_\eta(t) C_\eta^T(t) R_\eta^{-1}(t) C_\eta(t) R_\eta^{-1}(t) P_\eta(t) \Big] P_\eta^{-1}(t) e_{\hat{\eta}} \\
&= -e_{\hat{\eta}}^T \Pi e_{\hat{\eta}}
\end{aligned}
$$

where $\Pi = P_\eta^{-1}(t) \Big[ Q_\eta(t) + P_\eta(t) C_\eta^T(t) R_\eta^{-1}(t) C_\eta(t) R_\eta^{-1}(t) P_\eta(t) \Big] P_\eta^{-1}(t)$ and $\Pi$ is symmetric and positively defined.

For error system (12), the design controller should be $u = \psi(\chi, \hat{\eta})$. Considering the form of (12), constructing a controller as

$$
u = \psi(\chi, \hat{\eta}) = \frac{1}{G(\chi, \hat{\eta})} \big( -F(\chi, \hat{\eta}) + K e' + y^{d(r)} + u_s + u_h \big) \tag{13}
$$

with $e' = \begin{bmatrix} e_\chi^T, & e_{\eta^d}^T \end{bmatrix}^T = \begin{bmatrix} \left(\chi - \chi^d\right)^T, & \left(\hat{\eta} - \eta^d\right)^T \end{bmatrix}^T$, then the designed controller can be rewritten as

$$
\begin{aligned}
\dot{e} &= Ae + B\Big[ F(x) + (G(x) - G(\xi, \hat{\eta}) + G(\xi, \hat{\eta})) u - y^{d(r)} \Big] \\
&= Ae + B[\Delta F + \Delta G u + K e' + u_s + u_h] \\
&= (A + BK)e + B[\Delta F + \Delta G u - K e'' + u_s + u_h] \\
&= (A + BK)e + B M_F(\xi, \hat{\eta}) + B\big( \Delta G G^{-1}(\xi, \hat{\eta}) + I \big)(u_s + u_h) \Delta F
\end{aligned}
$$

where $\Delta F = F(x) - F(\xi, \hat{\eta}), \Delta G = G(x) - G(\xi, \hat{\eta}), e'' = \begin{bmatrix} 0, & e_{\hat{\eta}}^T \end{bmatrix}^T$, $M_F(\xi, \hat{\eta}) = \big( \Delta F + \Delta G G^{-1}(\xi, \hat{\eta}) \big)\big( -F(\xi, \hat{\eta}) + K e' + y^{d(r)} \big) - K e''$.

The following Assumptions are needed:

**Assumption 1.** *The expression of $\Delta G G^{-1}(\xi, \hat{\eta})$ is unknown, but it is bounded, so $\|\Delta G G^{-1}(\xi, \hat{\eta})\| \leq \kappa_G < 1$ is hold, where $\kappa_G$ is a known constant.*

**Assumption 2.** *The expression of $M_F(\xi, \hat{\eta})$ is unknown and is bounded, so $\|M_F(\xi, \hat{\eta})\| \leq (\rho_0 + \rho_1 \|e'\|) \|B^T P e'\|$ is hold, where $\rho_0$ and $\rho_1$ are unknown scalars.*

Then, for the closed loop system, the following Theorem is obtained:

**Theorem 1.** *For the state-tracking system (9) together with the estimation error system (12), under Assumption 1 and 2, if there exist matrices $P_e > 0$, and $K$, satisfying*

$$
P_e(A + BK) + (A + BK)^T P_e + P_e B R_e^{-1} B^T P_e + Q_e < 0 \tag{14}
$$

$$
\begin{bmatrix} -2\sqrt{Q_e \Pi} & P_e B \\ B^T P_e & -R_e^{-1} \end{bmatrix} < 0 \tag{15}
$$

where $R_e$ and $Q_e$ are given positive definite control matrices, then the proposed adaptive controller (13) can guarantee the stability of (9) and the estimation error of (12) with

$$u_h = \frac{1}{2(1+\kappa_G)} R_e^{-1} B^T P_e e', \tag{16}$$

$$u_s = -\frac{\hat{\rho}_0 + \hat{\rho}_1 \|e'\|}{1+\kappa_G} sgn(B^T P_e e') \tag{17}$$

The adaptive law for $\hat{\rho}_0$, $\hat{\rho}_1$ is

$$\dot{\hat{\rho}}_0 = q_0 \|B^T P_e e'\|, \dot{\hat{\rho}}_1 = q_1 \|e\| \|B^T P_e e'\| \tag{18}$$

**Proof of Theorem 1.** Choosing the Lyapunov function as

$$V_e = \frac{1}{2} e^T P_e e + \frac{1}{2q_0} \tilde{\rho}_0^2 + \frac{1}{2q_1} \tilde{\rho}_1^2$$

where $q_0$ and $q_1$ are given scalars. Notice that $\dot{\tilde{\rho}}_0 = -\dot{\hat{\rho}}_0$ and $\dot{\tilde{\rho}}_1 = -\dot{\hat{\rho}}_1$, differentiating (12)

$$\begin{aligned}
\dot{V}_e &= \frac{1}{2}\dot{e}^T P_e e + \frac{1}{2}e^T P_e \dot{e} - \%\frac{1}{q_0}\tilde{\rho}_0^T \dot{\hat{\rho}}_0 - \frac{1}{q_1}\tilde{\rho}_1^T \dot{\hat{\rho}}_1 \\
&= e^T[A+BK]^T P_e e + u_h^T (I + \Delta G G^{-1}(\xi, \hat{\eta}))^T B^T P_e e \\
&\quad + u_s^T (I + \Delta G G^{-1}(\xi, \hat{\eta}))^T B^T P_e e + M_F^T(\xi, \hat{\eta}) B^T P_e e \\
&\quad - \frac{1}{q_0}\tilde{\rho}_0^T \dot{\hat{\rho}}_0 - \frac{1}{q_1}\tilde{\rho}_1^T \dot{\hat{\rho}}_1
\end{aligned}$$

since $\|\Delta G G^{-1}(x|\theta_g)\| \le \kappa_G < 1, u_h = \frac{1}{2(1+\kappa_G)} R_e^{-1} B^T P_e e'$

$$u_h^T (I + \Delta G G^{-1}(x|\theta_g))^T B^T P_e e \le \frac{1}{2} e'^T P_e B R_e^{-1} B^T P_e e \le \frac{1}{2}(e - e'')^T P_e B R_e^{-1} B^T P_e e$$

For $M_F(x)$, $B^T P_e e \|e\| \le a\|e'\|$, and $u_s = -\frac{\hat{\rho}_0 + \hat{\rho}_1 \|e'\|}{(1+\kappa_G)} sgn(B^T P_e e'), \dot{\hat{\rho}}_0 = q_0\|B^T P_e e'\|, \dot{\hat{\rho}}_1 = q_1\|e\|\|B^T P_e e'\|$, then

$$\begin{aligned}
&u_s^T (I + \Delta G G^{-1}(x|\theta_g))^T B^T P_e e + M_F(x) B^T P_e e \\
&\le (\rho_0 - \hat{\rho}_0)\|B^T P_e e'\| - \frac{1}{q_0}\dot{\hat{\rho}}_0\tilde{\rho}_0 + (\rho_1\|e'\| - \hat{\rho}_1\|e'\|)\|B^T P_e e'\| - \frac{1}{q_1}\dot{\hat{\rho}}_1\tilde{\rho}_1 \\
&\le 0
\end{aligned}$$

Then,

$$\begin{aligned}
\dot{V}_e &\le \frac{1}{2}e^T[P_e(A+BK) + (A+BK)P_e]e + \frac{1}{2}e^T P_e B R_e^{-1} B^T P_e e - \frac{1}{2}e''^T P_e B R_e^{-1} B^T P_e e \\
&\le -\frac{1}{2}e^T Q_e e - \frac{1}{2}e''^T P_e B R_e^{-1} B^T P_e e
\end{aligned}$$

Since $e'' = \begin{bmatrix} 0, & e_{\hat{\eta}}^T \end{bmatrix}^T, \Lambda + P_e B R_e^{-1} B^T P_e < -Q_e$

$$\begin{aligned}
2\dot{V}_e &\le -\begin{bmatrix} e & e''^T \end{bmatrix} \begin{bmatrix} Q_e & \frac{P_e B R_e^{-1} B^T P_e}{2} \\ \frac{P_e B R_e^{-1} B^T P_e}{2} & \Pi \end{bmatrix} \begin{bmatrix} e \\ e'' \end{bmatrix} \\
&\le -\begin{bmatrix} e & e''^T \end{bmatrix} F \begin{bmatrix} e \\ e'' \end{bmatrix}
\end{aligned}$$

where $F = \begin{bmatrix} Q_e & \frac{P_e B R_e^{-1} B^T P_e}{2} \\ \frac{P_e B R_e^{-1} B^T P_e}{2} & \Pi \end{bmatrix}$. Since $Q_e > 0, \Pi > 0, (P_e B R_e^{-1} B^T P_e)^2 < 4Q_e\Pi$, then $F$ is positively defined. Then,

$$\dot{V}_e \le 0$$

Then, the proof is completed. □

## 5. Simulation Results

A numerical simulation is proposed for the flexible-state estimation method and the-tracking-controller-design method given in this paper. The trim point of HFV can be found in [17], *A* and *B* in (12) can then be computed with

$$
A = \begin{bmatrix}
0 & 1 & 0 & 0 & 0 \\
0 & 0 & 0 & 0 & 0 \\
0 & 0 & 0 & 0 & 0 \\
7.073 \times 10^{-8} & 1.388 \times 10^{-8} & 7.186 \times 10^{-7} & 0.0696 & 1 \\
-2.29 \times 10^{-5} & 2.23 \times 10^{-5} & 1.27 \times 10^{-4} & 11.63 & 0 \\
0 & 0 & 0 & 0 & 0 \\
0 & 0 & 0 & 4647.82 & 0 \\
0 & 0 & 0 & 0 & 0 \\
0.021 & -1.555 \times 10^{-3} & 0.1176 & -5100.67 & 0
\end{bmatrix}
$$

$$
\begin{bmatrix}
0 & 0 & 0 & 0 \\
0 & 0 & 0 & 0 \\
0 & 0 & 0 & 0 \\
0 & 0 & 0 & 0 \\
0.0745 & -1.80 \times 10^{-4} & 0.263 & -5.26 \times 10^{-4} \\
0 & 1 & 0 & 0 \\
-272.25 & 0.66 & 7.207 \times 10^{-5} & 1.44 \times 10^{-7} \\
0 & 0 & 0 & 1 \\
4.90 \times 10^{-5} & 1.189 \times 10^{-7} & -400 & 0.8
\end{bmatrix}
$$

$$
B = \begin{bmatrix}
0 & 0 \\
1 & 0 \\
0 & 1 \\
1.7 \times 10^{-6} & -3.39 \times 10^{-6} \\
-0.0161 & -0.0054 \\
0 & 0 \\
-3.35 \times 10^{-5} & 1.8304 \times 10^{-4} \\
0 & 0 \\
18.0596 & -1.24
\end{bmatrix}.
$$

For the application of EKF, the initial states are $\hat{\eta}(0) = \begin{bmatrix} 1.5122 & 0 & 1.122 & 0 \end{bmatrix}^{T}$, $Q_{\eta}(0) = 1 \times 10^{-5} I$, $R_{\eta}(0) = diag(10, 10)$. Then, $K$ can be obtained and

$$
K = \begin{bmatrix}
-2.919 & -4.14 & 2.65 & -2771.93 \\
0.848 & 0.9295 & 4.629 & 386.93\% \\
-822.22 & 0.0153 & 0.0017 & 0.5 & 0.0526 \\
114.79 & -0.0033 & -0.00027 & -0.05 & 0.00013\%
\end{bmatrix}.
$$

The setting of the simulation, such as the specific value of the reference command, and the setting of the instruction filter, are all the same as in [17]. The proposed observer-based controller is based on the nonlinear model of HFV. The reference command tracking performance can be found in Figure 2. As shown in Figure 2, the tracking error was almost zero for both the attitude command and the velocity command.

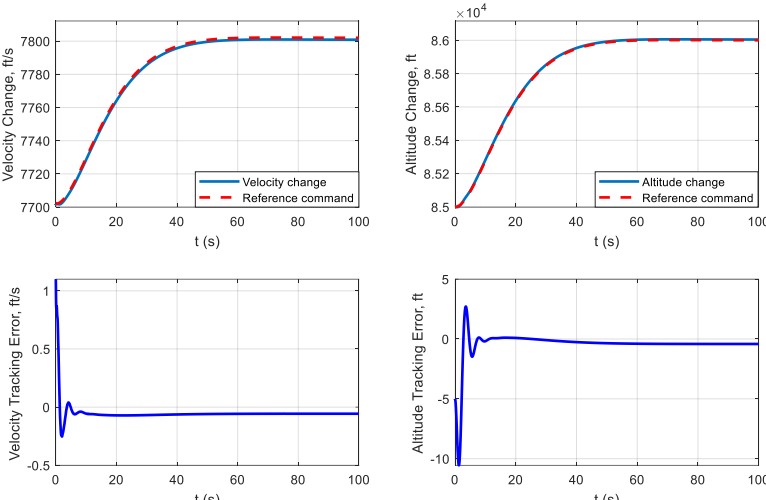

**Figure 2.** Tracking performance.

The control input of the proposed controller is given in Figure 3. As shown in Figure 3, the input of the plant was smooth and was within the limits of the control constraints; thus, the control input generated by the proposed controller is reasonable.

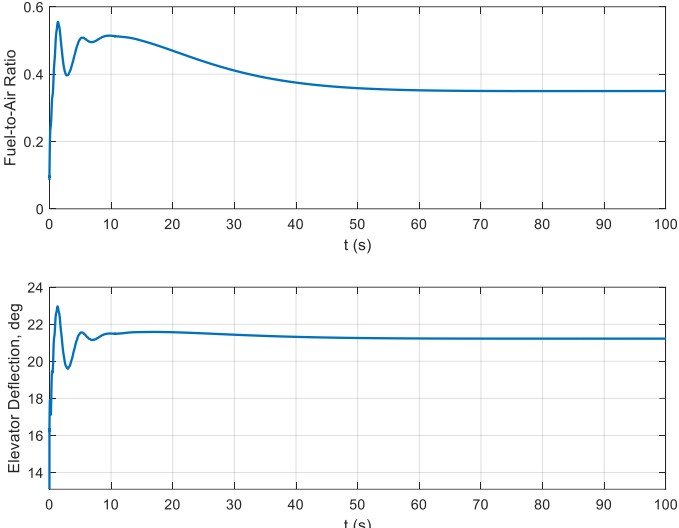

**Figure 3.** Input of the plant.

The estimated states and the real states are given in Figures 4–7, where Figures 4 and 5 are the first-order vibration of the HFV, $\eta_1$ and $\eta_2$, and Figures 6 and 7 are the second-order vibration of the HFV, $\dot{\eta}_1$ and $\dot{\eta}_2$, together with the observer error. From Figures 4–7, we can see that, after about 10 s, the observer errors all converged to zero; thus, the proposed EKF observation can obtain a good observer performance.

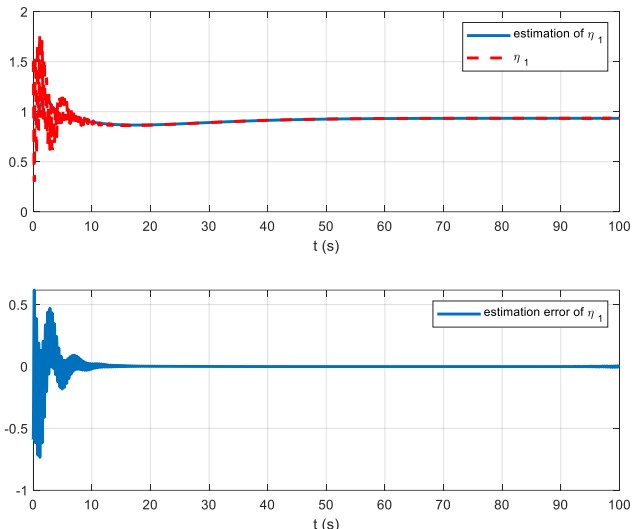

**Figure 4.** $\eta_1$ and its estimation.

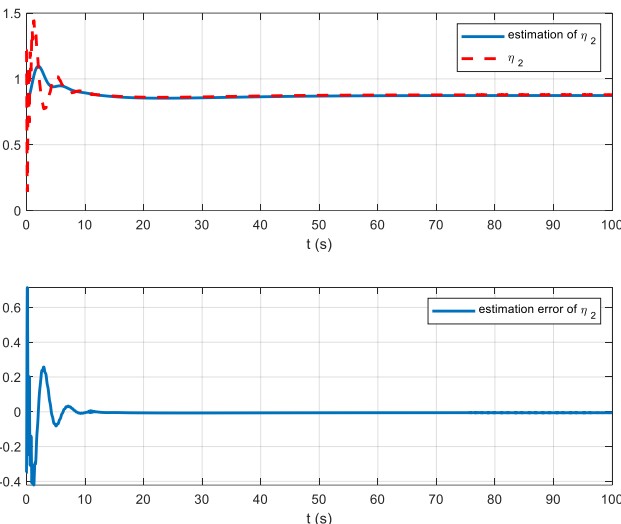

**Figure 5.** $\eta_2$ and its estimation.

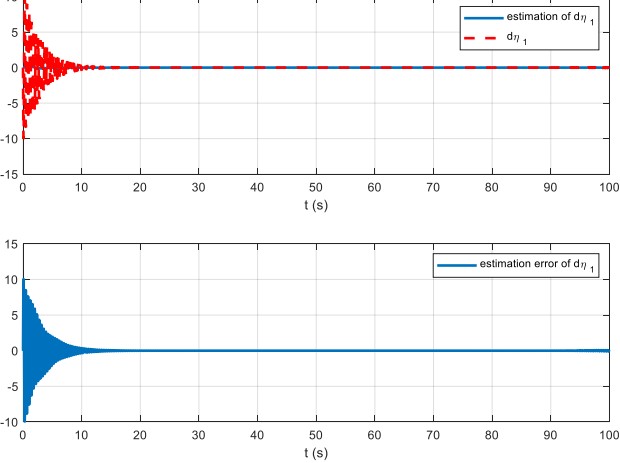

**Figure 6.** $\dot{\eta}_1$ and its estimation.

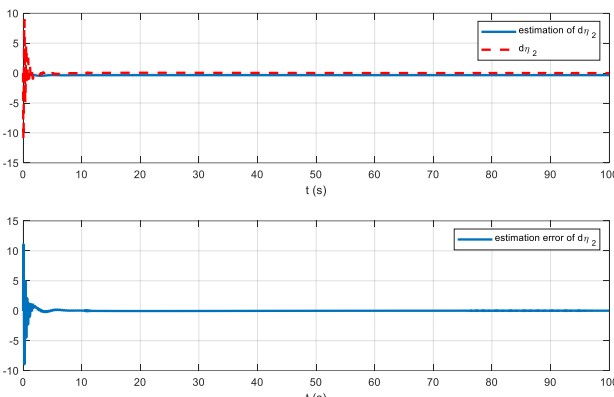

**Figure 7.** $\dot{\eta}_2$ and its estimation.

## 6. Conclusions

An angular-acceleration-based flexible-state estimation method for HFVs was proposed, together with an output-tracking-controller-design method for HFVs. By adding angular acceleration and designing an EKF observer, the flexible states of HFV could be estimated online. Then, by means of the nonlinear stable inversion technique and the estimation of flexible states, a nonlinear controller was constructed, and the stability of the proposed method was proven via the Lyapunov function. Finally, simulation results for HFVs were given to test the effectiveness of the proposed method. The proposed method provides an effective way to achieve the flexible-state estimation and observer-based controller design of HFVs. Using the proposed method, multiple elastic modal sensors can be replaced by an angular acceleration, so the manufacturing costs of HFVs are reduced and the controller performance is simultaneously guaranteed.

For the real application of the proposed method, disturbances in actual operations must be considered. In this case, further work is needed to test the applicability of the proposed method in real applications; for example, its efficacy under the influence of impulse disturbance, step disturbance, or modeling variations, etc.; with the obtaining of further testing results, the controller should be improved continuously.

**Author Contributions:** Conceptualization, D.Z. and X.H.; methodology, X.H.; software, X.Z.; validation, Y.X.; formal analysis, D.Z.; resources, Y.X. All authors have read and agreed to the published version of the manuscript.

**Funding:** This research was funded by the National Natural Science Foundation of China, grant number 62073265.

**Institutional Review Board Statement:** Not applicable.

**Informed Consent Statement:** Not applicable.

**Data Availability Statement:** All data used during the study appear in the submitted article.

**Conflicts of Interest:** The authors declare no conflict of interest.

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
