# Peer review of "Angular-Accelerometer-Based Flexible-State Estimation and Tracking Controller Design for Hypersonic Flight Vehicle"

_aerospace, doi:10.3390/aerospace9040206_

Round 1
Reviewer 1 Report
Overall, this is an interesting paper. I recommend following major changes.
1. Enrich the literature review in your work.
2. Highlight the innovative contributions of the proposed methodology in the Introduction Section.
3. Slightly elaborate the discussion related to the tracking controller design.
5. Comment on the originality / novelty of the said technique.
6. Can this technique be extended and applied to other electro-mechanical under-actuated systems?
7. Introduce additional test cases to better evaluate the proposed controller's efficacy; For example, I would suggest to test it under the influence of impulse disturbance, step disturbance, or modeling variations, etc.
8. Include a concise quantitative and qualitative comparative analysis of the experimental results.
9. Check for grammatical and spelling mistakes (if any).
Author Response
In this revision, all the comments from the reviewer(s) have been carefully taken into account and thoroughly implemented. Now, we would like to submit the revised version to be considered for possible publication in”Aerospace”.

Reviewer 2 Report
Angular Accelerometer Based Flexible States Estimation and Tracking Controller Design for Hypersonic Flight Vehicle
A brief summary
The paper proposes an angular accelerometer-based flexible state estimating method for hypersonic flight vehicles. By adding a pitch angel angular accelerometer and designing an Extended Kalman Filter based online estimating method, the flexible states of the flight vehicle can be got in real time.
Comments
- The idea of the research is interesting and presents enough novelty.
- The paper should attract an audience in the field of flight vehicle design.
- The paper fits the topics of the journal.
- The proposed method seems to be innovative and contains well-known hints of originality.
Weakness of the paper:
- Authors should anticipate some results in the abstract to capture the readers’ curiosity.
- The results of the paper should be better highlighted.
- The paper should be revised paying attention to the position of the figures.
- The most significant numerical results should also be reported in the conclusions.
- It is preferable to use impersonal style (i.e., avoiding I and we).
- Abbreviations should be defined when first used.
- Insert a drawing or a figure that helps the reader to understand the symbols used in the equations.
- Better clarify the positioning of the proposed angular accelerometer considering the vehicle.
- On page 13 the sentence "... then K in (??) can be got and ..." should be revised because it is incomplete
- The phrase "the tracking error is reasonable small" cannot be accepted in a scientific article. The concept of "small" or "great" number or difference is clearly related to the phenomenon under consideration.
- Figures from (1) to (6) should be anticipated in a specific section where they can be discussed in depth.
- The bibliography is too directed towards authors of Chinese nationality. Not that this is an absolute problem, but the contribution of previous knowledge to the research presented should be balanced.
- The journal format distributed by the Editor was not adopted in the article by the authors.
- … no more weaknesses!
The overall merit of presented research works and findings can be taken into consideration for publishing after incorporating the above suggestions.
Author Response

(The authors gave the same response as above.)

Round 2
Reviewer 1 Report
Well-revised. Good luck !
Author Response
Thank you for your recognition and support.
Reviewer 2 Report
The paper has improved and is almost ready for publication. I invite the authors to modify:
- abstract: adding a sentence with the main results obtained
- conclusions: better highlight the results obtained from the research by demonstrating how the proposed method adds or improves something compared to the methods currently in use.
Author Response
Thank you for your recognition and support.
In this revision, all the comments from the reviewer(s) have been carefully taken into account and thoroughly implemented. Now, we would like to submit the revised version to be considered for possible publication in”Aerospace”.
